# Impact of Liquidity Coverage Ratio on Performance of Select Indian Banks

Anureet Virk Sidhu [1], Shailesh Rastogi [1], Rajani Gupte [2] and Venkata Mrudula Bhimavarapu [1,*]

1   Symbiosis Institute of Business Management, Symbiosis International (Deemed University), Pune 412115, India; anureetv.virk@gmail.com (A.V.S.); krishnasgdas@gmail.com (S.R.)
2   Department of Management, Symbiosis International (Deemed University), Pune 412115, India; rajani-gupte@siu.edu.in
*   Correspondence: mrudulabhimavarapu@gmail.com

**Abstract:** The post-crisis liquidity framework improves banking stability by imposing stricter liquidity requirements. However, consistent bank performance continues to be an essential factor in achieving this goal. This study examines the impact of the liquidity coverage ratio (LCR) on the profitability and non-performing assets (NPAs) of Indian banks using annual data from 2010 to 2019. By applying the dynamic panel data regression technique, we found that compliance with the minimum level of the LCR reduces the net interest margins (NIMs) of banks due to a narrower interest spread, thereby impacting banks profitability. Moreover, the NPAs of the banks tend to grow with an increase in LCR. The study's findings have far-reaching implications for policymakers. Indian policymakers/regulators need to understand the strategies used by banks to meet liquidity standards and, if necessary, revisit the policy framework to achieve better compliance results. The study's framework establishes a foundation that can be used for conducting similar research in other complex geographies such as India.

**Keywords:** LCR; performance; NPAs; NIMs; liquidity; dynamic panel data analysis

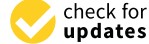



## 1. Introduction

In any economy, the central function of banks is financial intermediation, which makes them intrinsically susceptible to liquidity risk of both an institutional and a market-specific nature. Maintaining a balanced level of liquidity is critical for banks from both an economic and an individual entity standpoint. Evidence of the importance of liquidity for banks and the economy can be found in the past. For example, in the initial *liquidity phase* of the financial crisis that started in 2007, many banks faced difficulties despite acceptable capital levels as their liquidity was being managed in an imprudent manner. In December 2008, for instance, when the U.S. government bailed out Citibank, the bank's proportion of capital to risk-weighted assets still surpassed 11% (Kowalik 2013).

Another essential element related to the crisis was a sudden fall in the availability of short-term funding in capital markets, which made things worse for the already struggling banks (BCBS (BCBS) 2013; Banerjee and Mio 2018). The crisis thus highlighted the vital role that liquidity needs to perform in the optimum operation of both the banking sector and financial markets.

To address the weaknesses exposed by the crisis, BCBS rolled out Basel III: "International framework for liquidity risk measurement, standards, and monitoring" in December 2010 (BCBS 2010). The regulation aims to control liquidity through two new liquidity ratios, the Liquidity Coverage Ratio (LCR) and the Net Stable Funding Ratio (NSFR). The LCR addresses liquidity risk and requires banks to maintain adequate high-quality liquid assets (HQLAs) stock relative to projected short-term flows. The NSFR addresses funding risk and promotes long-term bank stability by prompting banks to adopt safer and more stable funding sources.

This study investigates the relationship between liquidity creation and bank performance in light of the aforementioned regulatory changes. We want to answer specific questions about the effect of liquidity requirements on banks, such as how short-term liquidity affects bank profitability. What effect do stricter liquidity standards have on the non-performing assets (NPAs) of banks? More specifically, will the increased liquidity alleviate or exacerbate a bank's NPA problem? The answers to these questions may have critical policy implications for the banking system's health and soundness.

## 1.1. Liquidity Regulations and Bank Performance

There are differing perspectives on how these regulations affect bank performance. According to one point of view, new liquidity rules would increase bank profitability and, as a result, the stability and resilience of the banking sector. Another point contends that banks may initially earn higher profits while adhering to liquidity standards but then lose money. The persistent losses of banks have the potential to destabilize the entire economic system. As a result, the goal of improving financial resilience through liquidity ratios is defeated. Thus, examining how the new liquidity regime affects key bank performance parameters is critical.

On the subject of liquidity regulations, researchers adhere to various schools of thought. On the one hand, Khan et al. (2015) maintain that the argument that tighter liquidity rules will reduce bank profitability is flawed. According to the authors, increased liquidity reduces the likelihood of default, which lowers the bank's financing costs and results in higher profits. Improved bank performance also translates into broader comprehensive macroeconomic benefits through lower bankruptcy risks (Konovalova 2016) and lower social costs (Stein 2013), improving the economy's overall stability (Papadamou et al. 2021).

On the other hand, the banking industry claims that stricter liquidity rules would limit a bank's ability to create liquidity and thus reduce profitability. The industry viewpoint is consistent with King's (2013) findings, which suggest that increasing liquidity ratios have a negative impact on bank performance. Bordeleau and Graham (2010) further observe that increased bank profitability due to higher liquidity is not an everlasting phenomenon. Beyond a point, holding more of these highly liquid assets diminishes profits for banks as the opportunity cost of holding additional units of liquid assets outweighs the benefits of lower default risk.

The above discussion highlights that the new regulation has many merits in the short term and suggests a convincing case for ring-fencing sound liquidity practices. Nevertheless, the long-term perspective brings forth the argument of trading off long-term performance deterioration against short-term compliance.

## 1.2. Liquidity Regulations and NPAs

To comply with liquidity regulations, banks can use a variety of strategies. Existing research (Hoerova et al. 2018; Polizzi et al. 2020) shows that higher liquidity requirements restrain risk-taking behavior by banks and aid in the development of a more reliable asset base. As a result, compliance has taken precedence over profit-making for banks. This shift in priorities results in a higher quality asset base in the short term, but additional research is required to assess the long-term impact due to the behavioral characteristics involved.

## 1.3. Liquidity Regulations and Institutional Environment in India

Regional demographics are significant in deciding the liquidity impact. Other factors such as enterprise size, asset quality, maturity period, and industry competitiveness complicate the study of the relationship between liquidity and bank performance even further. Given the size, volume, and variety of enterprises in Indian geography, it is critical to investigate how bank performance behaves under the new rules. This research will aid in the establishment of a standard framework for other regions with comparable economic conditions.

We find, however, that little research is available on the topic regarding India. Furthermore, whatever studies exist, they pertain to the pre-implementation period, during which the LCR was calculated using a rather simplistic, straightforward approach that disregards even some of the essential guidelines outlined in regulations. This necessitates a contextualized, in-depth investigation. This study fills the gap, as stated earlier, by examining the effect of liquidity regulations on the performance of Indian banks using data from 2010 to 2019, spanning both the pre-and post-implementation periods.

Further, one of the glaring gaps identified by the study is that most of the studies conducted so far assess the impact of NPAs on bank performance. However, in light of liquidity regulations, the impact, as mentioned earlier, was not assessed. This is pertinent for Indian banks as they have always struggled with the problem of higher NPAs. This study opens the doors to the Liquidity and NPA discussion and, at the same time assesses the impactof liquidity regulations on a bank's profitability.

In light of the problems highlighted above, this study aims to analyze the following:

- The impact of liquidity regulations on the performance of Indian banks with particular reference to short-term liquidity;
- The impact of liquidity regulations on NPA levels of Indian banks;

The study results, will help the policymakers understand how far the regulations have successfully achieved their goals regarding the Indian economy and whether additional steps are needed to make the impact more meaningful.

The remainder of this study consists of the following sections. Section 2 summarizes the Literature Review and develops the hypotheses. Section 3 describes the data, variables used in the study, and model applied for empirical analysis. Section 4 presents the observed results. Finally, Section 5 discusses the results and their implications, and Section 6 draws the conclusion.

## 2. Literature Review and Hypothesis Development

### 2.1. Need and Rationale for Liquidity Regulations

In the current global economic scenario, having robust governing regulation around bank liquidity is imperative for better economic conditions. The primary factor these regulations try to address is Market Failure arising due to asymmetric information and moral hazards. In this section, we review these in detail.

### 2.1.1. Market Failures

Rochet (2008) advocated the need for regulations around bank liquidity by explaining the market failures that arise because of asymmetric information and moral hazard. The discussion on the same is below.

### 2.1.2. Asymmetric Information

Diamond and Dybvig (1983) discuss two vital roles banks perform in an economy. First, banks boost economic welfare by improving production efficiency. This is achieved by channeling short-term investor assets into higher-return assets (longer duration). Second, banks help the depositors by facilitating immediate fulfillment of their unanticipated liquidity needs via pooled depositor funds. However, they suggest that in an economy where the entire wealth mediates through a banking sector, runs can be prompted through panics shaped by uninformed depositor expectations about premature withdrawals or by adverse signals about a bank default risk.

Diamond and Kashyap (2016) extend Diamond and Dybvig's (1983) framework and argue that since depositors cannot establish bank liquidity holdings, banks, in their endeavor to maximize profits, give precedence to lending rather than holding liquid assets, making themselves vulnerable to bank runs. Therefore, they favor having a regulatory liquidity mandate which would curtail excessive riskier lending by banks and prevent bank runs. Further, Rochet (2008) and Calomiris and Kahn (1991) also advocate that by

holding sufficient liquid reserves, banks can guard themselves against the liquidity risk arising from a few early "misinformed" withdrawals and avoid the crisis.

The above literature suggests that a market perception-based scenario, which can seriously destroy a bank's liquidity cushion, arises when customers act under limited market information and start retrieving funds and deposits, thereby causing a severe liquidity crunch for a bank. This implies that one of the critical needs for liquidity is to uphold a bank's position in the eyes of its investors/customers and the industry in which it operates. Significant liquidity buffers make institutions less susceptible to runs, as they enhance creditor confidence in the institutions' ability to service their obligations (Kowalik 2013).

### 2.1.3. Moral Hazard

In the current global economy, the financial ecosystem observes deep interconnectedness, wherein the failure of one of the participants leads to a connected failure across similar participants and results in a collective failure of the financial system globally. Empirical evidence suggests that regulators cannot afford not to arbitrate when the crisis is systemic (Acharya and Yorulmazer 2008). Hoggarth et al. (2004) studied the policies adopted by government while helping banks in need and concluded that the government embraced different strategies during different crisis situations. While providing aid to individual banks, the government never burdened the citizens. In comparison, the resolution strategies mainly involved government support during the systemic crisis, making it a prevalent norm in the banking community to depend on the central banks for the liquidity cushion.

However, the central bank's role of being the lender of last resort (LoLR) gives rise to the problem of moral hazard amongst the banks by incentivizing them to assume more risk. Rochet (2008) argued that anticipation by banks of government intervention prompts them to take on excessive exposure and suggested that liquidity requirements would deter such behavior and prevent market failures. Consistent with Rochet's argument, Stein (2013) advocated implementing liquidity regulation as a more promising solution than central banks acting as LoLR which encourages banks to be less prudent and proves to be socially expensive in a crisis.

The account of literature in this section indicates that different government aids can create moral hazard problems in banks, wherein banks are tempted to take on more risks in order to be profitable. This might give rise to market failures that would then run through the economies globally. However, the existence of a Liquidity regulation would restrict excessive risks taken by banks, saving the financial systems from distress.

### 2.2. Liquidity Ratios and Bank Profitability

The liquidity regulations aim to enhance bank capability to endure severe financial stress originating from either the financial system or the economy (BCBS 2013). Therefore, the incubation and design of the liquidity ratios are such that they should lead to a higher stock of liquid assets. Sufficient liquidity improves bank performance, reduces insolvency risk, and advances robustness and resiliency during intervals of stress.

Khan et al. (2015), in their study of U.S. commercial banks, observe that bank interest expenses reduce in response to the improved bank liquidity as fund providers are willing to supply funding at lower prices, which advances bank NIMs; thus, the effect of higher liquidity on NIMs is positive. Similarly, Mashamba (2018) examines the impact of liquidity regulations on eleven developing economies for 40 banks between 2011 to 2016. The study's empirical results report that regulatory pressure originating from the LCR improves bank profitability. The findings are consistent with those of Said (2014), who suggests that a bank's ability to better manage its stable funding sources and asset liquidity is an advantage, leading to better bank performance. Furthermore, Berger and Bouwman (2009) demonstrate that added net surpluses are distributed among stakeholders with higher liquidity creation, hence bank valuation is positively related to the liquidity ratios.

However, the positive relationship between liquidity and performance may not be permanent. Le et al. (2020) prove the existence of a quadratic relationship between liquidity ratios and bank profitability for U.S. commercial banks. They highlight that when banks increase their liquidity in the initial stages, they witness improved profitability. The increased profits can be attributed to the enhanced efficiency of banks as they become more effective in capital mobilization and allocation. However, Le et al. (2020) notice a turning point in this efficiency as the opportunity cost of holding these additional liquid assets with higher costs outweighs their relatively low return. These results are not different from previous studies (Bordeleau and Graham 2010; Tran et al. 2016), which concluded that if not taken care of, the cost of highly liquid assets can adversely impact a bank's resiliency. A plausible explanation for the gradual decrease in bank profitability can be found in the literature below.

In order to be compliant with the rules, banks tend to adjust the constitution of their assets toward HQLA (Banerjee and Mio 2018; Duijm and Wierts 2016; Fender and Lewrick 2013), leading to increased competition for the categories of funding considered preferable under the rule (Hartlage 2012). The enhanced demand results in higher asset prices and, consequently, increases bank financing costs. The increased prices further impact the yield of these HQLAs. Fuhrer et al. (2017) found that the differentiation in yield existed between Level 1, 2, and non HQLA securities prior to regulatory changes. Still, the introduction of the LCR has further widened this gap by bringing in the HQLA premium of 4 bps. Thus, the marginal cost of acquiring additional units of liquid assets starts exceeding the relative return.

The above discussion highlights that higher liquidity levels might initially improve bank profitability. However, beyond a point, holding more liquid assets starts having an adverse impact on the performance of banks, and bank profitability starts declining as more and more liquidity is infused.

**Hypothesis 1 (H1).** *LCR and bank profitability exhibit an inverted U-shaped relationship.*

### 2.3. Liquidity Ratios and NPAs

The financial well-being of the banking industry in any economy can be adjudged by its current level of non-performing assets; the lower, the better (Das and Dutta 2014). Studies conducted over different periods across the globe point out that substantially higher non-performing assets worsen bank performance and productivity, resulting in an economic slowdown (Berger and DeYoung 1997; De Bock and Demyanets 2012; Zhang et al. 2016; Kadioglu and Ocal 2017).

However, with the advent of liquidity regulations, banks tend to modify their lending behavior to comply with the rules. With regard to the same, Paulet (2018) finds that European financial institutions altered their credit distribution in favor of more stable and safer corporates following the implementation of stricter liquidity norms. Similarly, Hoerova et al. (2018) observe that liquidity requirements restrict bank's risk-taking practices, resulting in a sounder portfolio constitution. Consistent with the above studies, Polizzi et al. (2020) highlight that new regulations curb bank participation in non-traditional business activities, which lowers the overall risk level for banks.

The discussion above shows that while complying with the new liquidity ratios, banks curtail high-risk investments and make better credit distribution decisions. This enables them to build a sounder and more stable asset portfolio with lower default risk. However, it is crucial to understand that each economic system has its own set of characteristics that governs its relationship with the different banking regulations. The same holds true for Indian banks. Indian banks have always struggled with the problem of high NPA levels and thus often exhibit moral hazards and resort to riskier lending practices to improve their performance. The cushion of higher liquidity levels under the new regime might further encourage banks to lean towards riskier business activities to maximize profits,

exacerbating the NPA situation. It, therefore, becomes crucial to investigate how Indian bank NPAs respond to the new liquidity guidelines.

**Hypothesis 2 (H2).** *LCR has a significant impact on the NPAs of Indian banks.*

### 3. Data and Methodology

*3.1. Sources of Data*

For various reasons, including the 2008 global financial crisis, the Indian banking industry is undergoing major restructuring (D'Amato and Gallo 2019; Naheed et al. 2021; Khalil and Slimene 2021). Furthermore, the merger of government-induced public sector banks in 2019 caused significant upheaval in the Indian banking sector. As a result, bank data were collected until 2019. The data for the current study were gathered from the CMIE Prowess database and the official website of the Reserve Bank of India, which provides all the information related to the banking system in India (Al-Homaidi et al. 2018). Additional information was compiled from the various bank official websites and annual reports. Initially, 34 private and public sector banks in India were considered for the study due to the lack of availability of the data, and to fit the balanced panel of the study, three banks were dropped from the sample, and the final sample was fixed with thirty-one banks. For 31 Indian commercial banks, data were extracted from 2010 to 2019 for ten years.

Moreover, there are instances in the literature that justify using data for ten years in a study (Al-Homaidi et al. 2018). Profitability, measured using NIM (Tarus et al. 2012), returns estimated using ROA (Al Nimer et al. 2015), and risk, measured using NPA (Kiran and Jones 2016), are the variables used in the current study to measure the performance of the banks.

*3.2. Variables Considered for the Study*

Table 1 describes the most significant variables used for the analysis. LCR represents the liquidity coverage ratio that measures a bank's pliability over thirty days, if and when a financial crisis occurs (Hartlage 2012). LCR and LCR_P, i.e., a proxy for the LCR, are the study's exogenous variables. NIM (net interest margin) (Tarus et al. 2012), ROA (Return on assets) (Al Nimer et al. 2015), and NPA (Non-performing assets) (Kiran and Jones 2016) are the dependent variables of the dynamic models tested. A detailed description of the predicted variables is presented in Table 1.

**Table 1.** Variable Description.

| Variable Name | Symbol | Type | Description | Literature |
|---|---|---|---|---|
| Liquidity Coverage ratio | LCR | IV | Measures a bank's capability of fulfilling its obligations for thirty days if crisis circumstances arise. It is calculated as per BASEL guidelines by dividing the HQLAs by expected Net cash inflows | (Hartlage 2012) |
| Quadratic term of LCR | $LCR^2$ | IV | The squared term of LCR is obtained by multiplying (LCR*LCR) | (Boubakri et al. 2020) |
| Bank Liquidity | LCR_P | IV | The ratio is attained by dividing the Total loans by Total deposits. It explains how banks can fulfill their short-term requirement promptly. A higher ratio indicates that banks are not liquid enough, which, consequently, will increase risk and lower profitability | (Pak 2020) |
| Net Interest Margin | NIM | DV | The difference between interest earned on its advances and interest paid to its depositors. It is a profitability measure | (Tarus et al. 2012) |
| Return on Assets | ROA | DV | The ratio is attained by dividing the earnings before interest and tax by average total assets | (Al Nimer et al. 2015) |
| Non-Performing Assets | NPA | DV | An amount due for more than a period of 180 days from the date of borrowing | (Kiran and Jones 2016) |
| Equity Assets | Equity_assets | CV | It represents the net worth or the asset proportion of the bank | (Pak 2020) |
| Bank Size | Bank_size | CV | The size of the bank is determined by the total assets held by the respective bank | (Vo 2018) |

Source: Author compilation. Note: Table 1 describes the variables used for the study.

Furthermore, two control variables are also considered to have a good fit for the model. l_Banksize is the control variable and is taken as the natural log of the size of the bank. It is measured as the total assets held by the respective Bank (Vo 2018). Equity assets are another control variable representing the net worth or the asset proportion of the bank.

### 3.3. Conceptual Model of the Study

The conceptual model is designed for empirical testing. The model is proposed in Figure 1. It is evident from Figure 1 that the study aims to check the impact of the liquidity coverage ratio on the profitability and NPA of the banks. A bank's profitability (is established using the two different proxies of NIM and ROA) and NPA using the econometric Equations (1) and (2). The proposed model is tested using dynamic panel data regression to obtain the results discussed in detail in the following sections.

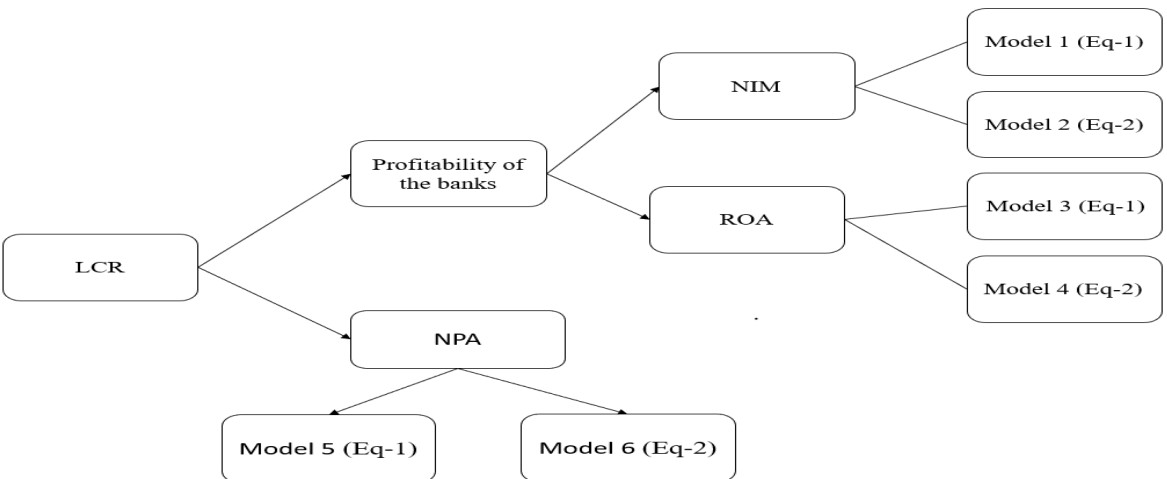

**Figure 1.** Conceptual Model of the study. Source: Author compilation. Note: Figure 1 explains the theoretical model adopted for the study.

### 3.4. Methodology

The data were analyzed using panel data, which is advantageous over traditional cross-section or time series analysis (Baltagi and Baltagi 2008). Panel data analysis can reveal significantly more information than a single time series or cross-section analysis (Hsiao 2007). Panel data can provide a clearer picture of regression analysis by exhibiting more information (Hsiao 2007; Wooldridge 2015). The authors used dynamic panel data regression due to the endogeneity problem in the study. A related key advantage of dynamic panel models is the ability to determine short and long-run coefficient values. Furthermore, such models allow researchers to select which explanatory variables are potentially endogenous or exogenous (Nickell 1981). The econometric model used for the analysis is presented below.

$$Y_{it} = \alpha + \beta_1 Y_{it-1} + \beta_2 LCR_{it} + \beta_3 LCR_{it-1} + \beta_4 LCR^2_{it} + \beta_5 equity\_asset_{it} + \beta_6 l\_banksize_{it} + u_{it} \tag{1}$$

$$Y_{it} = \alpha + \beta_1 Y_{it-1} + \beta_2 LCR\_P_{it} + \beta_3 LCR\_P_{it-1} + \beta_4 LCR\_P^2_{it} + \beta_5 equity\_asset_{it} + \beta_6 l\_banksize_{it} + u_{it} \tag{2}$$

Equations (1) and (2) present the exogenous variables' liquidity coverage ratio (LCR) and proxy [2](LCR_P) regression models. $Y_{it}$ is the dependent variable representing profitability (i.e., NIM, NPA, or ROA) of the i-th bank at t-th time. Each dependent variable is tested for the Equations (1) and (2). $\alpha$ is the constant term and $u_{it}$ is the error term in the models. Further, equity_asset and l_banksize are considered control variables with a good fit for the model. Bank size and Equity assets are included as control variables because they are deciding factors in evaluating bank economic importance across segments/groups and can, thus, interfere with performance measurement.

## 4. Results

The results section initially states the descriptive statistics and the correlation between the variables used for the present study. Results of the regression are discussed in detail, model-wise. Finally, tis section concludes with an explanation of tests for endogeneity.

### 4.1. Descriptive and Correlation Matrix

Table 2 reveals the correlation between the study variables and their descriptive statistics. The low mean values of NIM (2.674) and ROA (0.6141) indicate that the banks cannot maximize the utilization of assets to attain returns. The lower mean value of NET NPA (3.014) confirms fewer risks associated with the banks in India. All the pairs of variables have a significant correlation between them, and many of them are perfectly significant with a *p*-value of (0.000). ROA and NIM are positively correlated, indicating that both earnings constructs designate the same direction. The negative correlation between NET NPA–ROA and NIM–NET NPA indicates the inverse relationship between the variables, implying that increase in one variable results in a decrease in the other variable. This implies that as the NET NPA of the banks in India increases, it will affect the decrease of ROA and NIM (which means a decrease in the quality or performance of the bank). The negative correlation between l_Banksize and ROA, NIM, and Equity_asset in Indian banks is due to diseconomies of scale. The positive correlation of Equity_assets with ROA and NIM indicates that as a bank's return on assets and net interest margin increases, its proportion of net worth also increases.

**Table 2.** Descriptive Statistics and Correlation.

| Correlation Matrix | | | | | | | |
|:---:|:---:|:---:|:---:|:---:|:---:|:---:|:---:|
| | | | | | | **Mean** | **SD** |
| | **ROA** | **NETNPA** | **NIM** | **Equity_Asset** | **l_Bank Size** | | |
| **ROA** | 1 | | | | | 0.6141 | 0.8896 |
| **NETNPA** | −0.7288 * (0.0000) | 1 | | | | 3.014 | 2.795 |
| **NIM** | 0.6007 * (0.0000) | −0.5657 * (0.0000) | 1 | | | 2.674 | 0.6283 |
| **Equity_asset** | 0.4139 * (0.0000) | −0.4068 * (0.0000) | 0.5352 * (0.0000) | 1 | | 0.0741 | 0.0282 |
| **l_Bank size** | −0.1328 * (0.0194) | 0.3225 * (0.0000) | −0.1240 * (0.0290) | −0.2237 * (0.0001) | 1 | 3073 | 4469 |

Source: Author compilation. Note: Values in the correlation matrix are correlation coefficients. Values in parenthesis are *p*-values. * Significant at 5%.

Table 3 Presents the median value and quartile range (lower quartile and upper quartile) of the variables selected for the study. The median is helpful because it indicates where the center value in a dataset is located. When a distribution is skewed or has outliers, the median is more beneficial to calculate than the mean. The small variance values of the variables indicate that the numbers in the data set are not far from the mean or each other. Positive values of skewness of the variables NETNPA, NIM, and Equity_asset imply skewed right, meaning that the distribution's right tail is longer than the left; negative values of the variables ROA and l_Bank size indicate negatively skewed or skewed left. The positive skewness of the NETNPA highlights that a large percentage of sample banks have high NPA levels. This is primarily because of the moral hazard and risky lending practices that banks adopt for profit maximization. The high level of NPAs also explains why the Indian banks cannot generate sufficient returns from their assets and are incurring substantial losses. From Table 3, it is evident that the kurtosis of the Equity_asset indicates the heavy tail in the bell curve of the set and is considered to indicate an extreme value.

**Table 3.** Univariant Statistics of the study.

| Variable | No. of Observations | Median | Upper Quartile | Lower Quartile | Mean | Variance | Skewness | Kurtosis |
|---|---|---|---|---|---|---|---|---|
| **ROA** | 310 | 0.678 | 1.19 | 0.36 | 0.6141 | 0.7914 | −1.431 | 6.065 |
| **NETNPA** | 310 | 2.06 | 4.61 | 0.81 | 3.014 | 7.813 | 1.293 | 4.451 |
| **NIM** | 310 | 2.6 | 3.04 | 2.21 | 2.674 | 0.3947 | 0.4840 | 2.891 |
| **Equity_asset** | 310 | 0.0654 | 0.0884 | 0.0557 | 0.0741 | 0.0008 | 3.250 | 26.26 |
| **l_Bank size** | 310 | 12.17 | 12.84 | 10.99 | 3073 | 1.769 | −0.3768 | 2.752 |

Source: Author compilation. Note: Table 3 describes the Univariant Statistics for the current study.

*4.2. Dynamic Panel Data Results*

Compared with the results of regressions performed with OLS and fixed effects, System GMM is the preferred technique (Akbar et al. 2016). Over the last fifteen years, GMM estimation has emerged as an indispensable and unifying framework for interference in econometrics (Imbens 1997). GMM estimators were created by Arellano and Bover (1995) and Blundell and Bond (1998). The primary advantage of this model is that it incorporates more instrument variables. The lagged difference is introduced as an independent variable in the original equation, and the lagged values are introduced as independent variables in the differential equations. The current study employs dynamic panel data regression. Another significant virtue is the dynamic panel model's capacity to compute short and long-run coefficient values. Using a dynamic panel entails combining the lagged dependent variables with the exploratory variables.

Furthermore, the model is estimated using GMM, which works similarly, overcoming endogeneity issues. Using such models, researchers can choose which explanatory factors are potentially endogenous or exogenous. The regression analysis uses three dependent variables, NIM, ROA, NPA, and two exogenous variables (LCR and LCR_P). Models 1 and 2 analyze the association of IV (independent variable) with NIM, and Models 3 and 4 test the relation associated with the ROA. Similarly, Models 5 and 6 test for NET NPA. Due to endogeneity issues, the study adopted dynamic panel analysis (Wooldridge 2015).

For all the models examined, the perfect significant *p*-value for the A–B test (Arellano–Bond test) at lag1 (*p* < 0.000) eliminates the problem of autocorrelation (Baltagi and Baltagi 2008). Sufian and Habibullah (2010) recommend using the Arrelano–Bond A R (1) test at the 5% significance level. We can validate that our instruments are sufficiently statistically independent and that no first-order serial correlation exists. The Sargan test for overidentifying is significant for all the models with a *p*-value less than 0.05, which signifies no overidentification issue. The authors used Sargan's test to validate their findings. The significant lagged dependent variable coefficient confirms the model's dynamic nature. As a result, we can justify using the dynamic panel data model estimate (Sufian and Habibullah 2010).

Table 4 presents the outcomes of Model 1 and Model 2. From Table 4, it is evident that the previous NIM positively impacts present NIM (−1) with an absolute significant *p*-value [0.4039951 * (0.000)] in Model 1 and [0.3891872 * (0.001)] in Model 2. The LCR in Model 1 demonstrates that it holds an inverse impact on the NIM as it is negatively significant (−0.4101052, *p*-value 0.017 < 0.05), whereas LCR_P shows a significant positive (1.58785, *p*-value 0.034 < 0.05) relation with the NIM. However, neither the LCR (−1) nor LCR_P (−1) show any significance, as the coefficients are insignificant with *p*-values of (0.755) and (0.755), respectively, at a 5% significance level. Among the control variables considered for the study, l_banksize is positive in both models, whereas equity_asset is positively associated in only Model 1.

The significant LCR in Model 1 and LCR_P in Model 2 interpret a linear association between the variables, whereas the insignificant quadratic terms imply and show evidence that the variables have no quadratic association. The negative coefficient between LCR and NIM in Model 1 demonstrates that with an increase in the LCR levels, the NIM of banks decreases. Further, in Model 2, the positive coefficient of LCR_P with NIM implies that as the banks become less liquid, the NIM of banks increases. Since the LCR_P is calculated by

dividing total loans by total deposits, a higher ratio means that banks are holding more loans on their balance sheets than deposits. This explains why NIMs increase with an increase in LCR_P, as bank interest income from loans is more than the interest expense they need to bear on lower deposits. Another interpretation of the results for Model 2 is that as banks liquidity increases, i.e., when the LCR_P ratio is lower, the NIM of banks decreases as their interest expenses increase with higher bank deposit levels. Thus, Models 1 and 2 indicate that with an increase in bank liquidity, the NIMs of banks decrease.

**Table 4.** Dynamic Panel Data Model (Models 1 and 2).

| | DV: NIM | | | | | |
|---|---|---|---|---|---|---|
| | **Model 1** | | | **Model 2** | | |
| | Coefficient | Standard Error | *p*-Value | Coefficient | Standard Error | *p*-Value |
| NIM (−1) | 0.4039951 * | 0.4039951 | 0.000 | 0.3891872 * | 0.1153348 | 0.001 |
| **LCR** | **−0.4101052 *** | **0.1699658** | **0.017** | **–** | **–** | **–** |
| LCR (−1) | 0.048291 | 0.1545187 | 0.755 | – | – | – |
| **LCR²** | **0.0128379** | **0.1875941** | **0.945** | **–** | **–** | **–** |
| **LCR_P** | **–** | **–** | **–** | **1.58785 *** | **0.7436887** | **0.034** |
| LCR_P (−1) | – | – | – | −0.7384876 | 0.7735318 | 0.341 |
| **LCR_P²** | **–** | **–** | **–** | **1.298533** | **3.811142** | **0.734** |
| equity_asset | 3.793078 * | 1.344769 | 0.005 | 1.902104 | 1.558685 | 0.224 |
| l_bank_size | 0.1803045 * | 0.0452334 | 0.000 | 0.1851561 * | 0.0412864 | 0.000 |
| Arnello–Bond AR (1) | −3.91 * (0.0000) | | | −3.91 * (0.0000) | | |
| Sargan Test | 51.95 * (0.001) | | | 55.97 *(0.0000) | | |

Source: Author compilation. Note: Sargan test tests the over-identification issues beneath the GMM framework. The null hypothesis of the Sargan test is that the dynamic panel data model has no over-identification problem. The Arnello–Bond test was employed to look for serial autocorrelation in the first differenced error terms of the order. *p*-values are * significant at a 5% level of significance.

Given the inverse relationship between bank liquidity and net interest margin, it is evident that the bank's performance deteriorates when higher liquidity is required. The adverse impact on profitability, in the long run, will affect the behavior of banks towards maintaining a high liquidity surplus; hence there might be a need to deep dive into the long-term factors affected by liquidity regulations and amend regulations accordingly.

Models 3 and 4 with ROA as a dependent variable (DV) are presented in Table 5. It is evident from Table 5 that the existing ROA is positively influenced by the preceding ROA (−1) with a perfect significant value of (0.000) in both Models 3 and 4. Exogenous variable LCR reveals an insignificant coefficient with a *p*-value of (0.157 > 0.05), indicating no influence on ROA, whereas LCR_P depicts a positively significant (coefficient with a *p*-value of 0.011 < 0.05), implying constructive association with ROA. However, neither the LCR (−1) nor LCR_P (−1) show any significance, as the coefficients are insignificant with *p*-values of (0.867 > 0.05) and (0.268 > 0.05), respectively, at the 5% significance level. Amongst the control variables, l_banksize shows a negatively significant association in both the models. In contrast, equity_asset holds no association.

The significant LCR_P and quadratic term LCR_P² in Model 4 show evidence of the variables' linear and quadratic association. In contrast, the insignificant LCR and LCR² in Model 3 imply and show evidence that there exists neither a linear nor a quadratic association of the variables. The positive linear association between LCR_P measured by Total loans/Total Deposits and ROA indicates that as bank liquidity increases, the ROA decreases and as bank liquidity decreases, the ROA increases, implying that LCR_P has a negative relationship with bank profitability.

**Table 5.** Dynamic Panel Data Model (Models 3 and 4).

| | DV: ROA | | | | | |
|---|---|---|---|---|---|---|
| | **Model 3** | | | **Model 4** | | |
| | Coefficient | Standard Error | *p*-Value | Coefficient | Standard Error | *p*-Value |
| ROA (−1) | 0.7788722 * | 0.0920342 | 0.000 | 0.5826836 * | 0.1034741 | 0.000 |
| **LCR** | **−0.4819404** | **0.3392385** | **0.157** | **–** | **–** | **–** |
| LCR (−1) | −0.0537254 | 0.3207734 | 0.867 | – | – | – |
| **LCR²** | **−0.1884799** | **0.1735601** | **0.279** | **–** | **–** | **–** |
| **LCR_P** | **–** | **–** | **–** | **4.525751 *** | **1.759773** | **0.011** |
| LCR_P (−1) | – | – | – | 1.80073 | 1.62341 | 0.268 |
| **LCR_P²** | **–** | **–** | **–** | **−16.82013 *** | **7.221463** | **0.021** |
| equity_asset | 0.2561507 | 3.12437 | 0.935 | −1.864066 | 3.227412 | 0.564 |
| l_bank_size | −0.2478031 * | 0.1097369 | 0.025 | −0.3266999 * | 0.1013761 | 0.001 |
| Arnello–Bond AR (1) | | −8.52 * (0.000) | | | −8.00 * (0.000) | |
| Sargan Test | | 100.75 * (0.000) | | | 107.08 * (0.000) | |

Source: Author Compilation. Note: Sargan test tests the over-identification issues beneath the GMM framework. The null hypothesis of the Sargan test is that the dynamic panel data model has no over-identification problem. The Arnello–Bond test was employed to look for serial autocorrelation in the first differenced error terms of the order. *p*-values are * significant at a 5% level of significance.

The significant results for the quadratic term LCR_P² demonstrate that the relationship curve of LCR_P and ROA is in the shape of an inverted U. The result highlights that, initially, with more loans and less liquidity, banks might be able to earn higher returns; however, subsequently, as liquidity continues to decrease with the increase in the bank loan component, their ROA starts declining. This decline in returns can be attributed to a possible increase in the corresponding non-performing loans of the banks.

Table 6 presents the results for Models 5 and 6 with NPA as a dependent variable. Similar to the results for Models 1 to 4, the current NPA is impacted by that preceding it [NPA (−1)]. From the presented results in Table 6, it is evident that the LCR variable shows a positive association at 5% significance, whereas LCR (−1) displays a negative relation at 10% significance with NPA in Model 5. The variable LCR_P in Model 6 depicts a significant coefficient at 5% significance with NPA. In detail, LCR (−1) shows a significant association as the coefficients show significance with a *p*-value of (0.063) at a 10% significance level. In contrast, it does not offer any importance as the coefficients show insignificance with a *p*-value of (0.134) at a 5% or 10% significance level.

Regarding the control variables where l_banksize shows a positively significant association in both the models at 5% significance, equity_asset holds a positive association at 10% significance in Model 6. The significant LCR in Model 5 and LCR_P in Model 6 shows evidence of a linear association of the variables, whereas the insignificant squared terms imply and show evidence of no quadratic association of the variables. The positive relationship between LCR and NPAs in Model 5 represents that with an increase in the LCR, the NPAs of banks also increase. Similarly, the negative relationship between LCR_P and NPAs, implies that as liquidity increases, the NPA levels of Indian banks increase. This demonstrates that improved bank resilience through enhanced liquidity levels heightens the moral hazard of banks and other risky lending activities, leading to higher NPA levels.

The conclusive evidence from Models 5 and 6 underscores that the increase in liquidity also leads to an increase in NPA for the banks. NPAs are the measuring yardstick of any bank's performance and its repercussions on the economy. An increase in NPAs would lead to bank performance deterioration and, to a large extent, defeat the purpose of liquidity regulations.

**Table 6.** Dynamic Panel Data Model (Models 5 and 6).

| | DV: NPA | | | | | |
|---|---|---|---|---|---|---|
| | **Model 5** | | | **Model 6** | | |
| | Coefficient | Standard Error | *p*-Value | Coefficient | Standard Error | *p*-Value |
| NPA (−1) | 0.5172013 * | 0.0797831 | 0.000 | 0.3822689 * | 0.0803585 | 0.000 |
| **LCR** | **3.046514 *** | **0.9946143** | **0.002** | **–** | **–** | **–** |
| LCR (−1) | −1.73721 ** | 0.9302507 | 0.063 | – | – | – |
| **LCR$^2$** | **0.2483428** | **0.4963393** | **0.617** | **–** | **–** | **–** |
| **LCR_P** | **–** | **–** | **–** | **−15.65964 *** | **4.647991** | **0.001** |
| LCR_P (−1) | – | – | – | 6.835003 | 4.544969 | 0.134 |
| **LCR_P$^2$** | **–** | **–** | **–** | **15.48432** | **18.1913** | **0.395** |
| Equity_asset | 3.684094 | 9.211233 | 0.690 | 18.81853 ** | 10.47831 | 0.074 |
| l_bank_size | 0.8490845 * | 0.3518012 | 0.017 | 1.08797 * | 0.3116191 | 0.001 |
| Arnello–Bond AR (1) | −6.90 * (0.000) | | | −7.12 * (0.000) | | |
| Sargan Test | 80.13 * (0.009) | | | 114.17 * (0.000) | | |

Source: Author compilation. Note: Sargan test tests the over-identification issues beneath the GMM framework. The null hypothesis of the Sargan test is that the dynamic panel data model has no over-identification problem. The Arnello–Bond test was employed to look for serial autocorrelation in the first differenced error terms of the order of *p*-values being * significant at 5% significance level and ** significant at 10% significance.

### 4.3. Test for Endogeneity

Results for endogeneity are presented in Table 7—the test conducted for the exogenous variables LCR, equity_asset, and l_banksize for all the DVs. LCR and l_banksize are tested endogenous with significant Durbin–Wu–Hausman chi2 and F-test in the models where ROA and NPA are regressands. At the same time, Equity_asset is endogenous with response variable NIM. To treat the issue of endogeneity, authors ran ivregress-2sls using an instrument variable l3. of the same variable. Results are presented in Tables 3–5.

**Table 7.** Results for Endogeneity Testing.

| | DV: NIM | | |
|---|---|---|---|
| | LCR | Equity_Asset | l_Bank_Size |
| Endogeneity (Durbin–Wu–Hausman) chi2 | 0.934742 (0.3336) | 10.4356 (0.0012) | 0.213585 (0.6440) |
| Endogeneity (Durbin–Wu–Hausman) F-test | 0.917155 (0.3393) | 10.7103 (0.0012) | 0.208869 (0.6481) |
| | **DV: ROA** | | |
| | LCR | Equity_Asset | l_Bank_Size |
| Endogeneity (Durbin–Wu–Hausman) chi2 | 10.4959 (0.0012) | 0.646005 (0.4215) | 6.11555 (0.0134) |
| Endogeneity (Durbin–Wu–Hausman) F-test | 10.7752 (0.0012) | 0.633005 (0.4271) | 6.1479 (0.0139) |
| | **DV: NPA** | | |
| | LCR | Equity_Asset | l_Bank_Size |
| Endogeneity (Durbin–Wu–Hausman) chi2 | 19.2272 (0.0000) | 2.13596 (0.1439) | 6.76367 (0.0093) |
| Endogeneity (Durbin–Wu–Hausman) F-test | 20.6104 (0.0000) | 2.10749 (0.1481) | 6.82041 (0.0097) |

Note: Results of endogeneity tested are presented in Table 7.

## 5. Discussion and Implication of Results

### 5.1. Discussion of Results

The paper aims to analyze the impact of LCR on the performance of Indian banks by studying how it influences bank NIM, ROA, and NPAs. Since the variable LCR has been calculated as per the BASEL III guidelines, it is a contextually more credible measure than LCR_P.

The research findings show that LCR$^2$ has an insignificant relationship with NIMs and ROA. Therefore, the hypothesis that the LCR has an inverted U-shaped relationship with the performance of banks is rejected.

However, the study does find LCR to have a significant inverse relationship with NIMs for Indian banks. Moreover, since the study results indicate that LCR does not significantly relate to ROA, it can be concluded that the negative impact of LCR on bank performance is primarily due to the narrowing spread between interest income and interest expenses. The findings are in line with results from King (2013), who argued that liquidity regulations would add to bank interest expenses, decrease the NIMs and, consequently, impact banks performance.

It is a known industry practice/fact that high-quality liquid assets have always had colossal demand from various financial industry players for competing uses, be it for safe investment or collateralization. However, unfortunately, these assets have a limited supply and cannot keep up with this massive demand. The resulting supply–demand mismatch causes the asset prices to shoot up and the yields to go down (Carlson et al. 2015). The introduction of liquidity regulations further heightens the demand for these HQLAs, giving rise to a *liquidity premium* (Handorf 2014). Therefore, the increasing funding costs of these liquid assets, combined with their decreasing yields, lower the banks' NIMs. With the economic focus shifting to broader issues like sustainable finance, corporate or supply chain, banks will need their current profit engines to run smoothly (Tseng et al. 2021; Bui et al. 2020). Therefore, it is imperative that regulatory compliance support rather than hinder their progress.

Currently, there is a dearth of literature examining the impact of new liquidity regulations on bank NPAs. This study explores this relationship for Indian banks and observes that LCR is positively related to the NPAs, implying that with an increase in liquidity, the NPAs will grow. The research results support the hypothesis that LCR is significantly associated with the NPAs of the banks. However, the nature of the association between LCR and NPAs needs an explanation which is provided below.

Indian banks have persistently struggled with the problem of NPAs. Under the stress of managing these high levels of NPAs, banks tend to become susceptible to moral hazards, resort to riskier lending, and worsen the NPA situation (Koudstaal and van Wijnbergen 2012). With the implementation of the liquidity regulations, as the banks become compliant and increase their holdings of highly liquid assets, their resiliency and confidence to withstand shock are enhanced (Schmaltz et al. 2014; Bressan 2018). The enhanced perceived stability prompts banks to engage in risky advancing activities to maximize profits. The study observed the same for Indian banks, wherein the proportion of unsecured advances started witnessing an upward trend post LCR implementation in 2015. Thus, liquidity regulation augments existing risky lending practices by Indian banks, leading to higher NPAs.

### 5.2. Implications

This study brings the new-found results that continued weak performance may severely affect the banking sector's long-term stability through restricted progress in capitalization and lending. This could also lead to drastic consequences from a policymaking perspective. The higher liquidity holdings leading to lower NIMs would force banks to have recourse to more risky investments, leading to higher NPA levels. Therefore, national regulators must pay close attention to possible alterations in banking business models and risk profiles post-regulation implementation. Additionally, Indian policymakers can



assess and recommend hybrid growth-oriented strategies that meet business goals and compliance. Steps can be initiated to formulate policies that go hand-in-hand with the liquidity rules and help resolve the NPAs problem by the lending activities of governing banks.

The study advocates the vital role that bank managers need to play in the meaningful implementation of the regulations. Decision-makers must be extra careful about the policies/strategies adopted to comply with the liquidity standards. They must be mindful of the potential repercussions of these strategies on the broader economy and act accordingly.

The role of bank management is not only limited to decision-making. The rising NPAs of a bank, and its decreasing NIMs, can shake investor and depositor confidence in its well-being. This can further impact its business and financial health. Thus, the bankers/management must work actively towards handling the expectations of their investors and depositors and strengthen their trust in the bank.

The research showcases that one of the primary reasons for the decrease in the NIMs is the increased financing cost of the liquid assets due to the supply–demand mismatch. It is a call for action from market regulators to resolve this financial market gap.

## 6. Conclusions

Although the impact of liquidity regulations on the performance of banks has been examined in the literature, there are very few studies that explored this relationship in the context of India. As it is imperative to study the impact of these regulations in different countries/economies, this study investigated the impact of the new liquidity standards on the profitability and NPAs of Indian banks. We analyzed more than 30 banks over a ten-year timeframe. Six different models were created to analyze the correlations between significant attributes using various weighted combinations. Models developed were based on the leading methodologies of dynamic panel data regression.

The study results document that an increase in the LCR and its components increases the funding costs and this has a detrimental impact on the performance of banks. Thus, while increased liquidity reduces the liquidity risk that banks face, it comes at the expense of profitability. In terms of NPA, the empirical evidence suggests that NPAs tend to grow in response to higher liquidity levels, thus, further adding to the stress on bank profitability. However, the results are robust and clearly show a relationship between bank performance and liquidity regulations. The limitation of the study lies in the fact that these data do not consider external market factors that can influence performance.

Contextualizing this discussion to India, the study finds that, contrary to the regulations' founding principles, an increase in liquidity does not result in a corresponding decrease in NPAs. This should be considered input for the next set of regulations, as different countries may require a slightly different version of the regulations based on their respective economic conditions. The study's findings further the existing body of knowledge regarding India. The diversity and vitality of the Indian economy necessitate in-depth examination at all levels. The study investigates various influencing attributes that capture the essence of the liquidity impact under various scenarios. This results in a defined framework that correlates liquidity and bank performance across all levels and attributes. This balance of complex financial institutions in operating geography is not often attempted. As a result, this study lays the groundwork for frameworks for complex geographies similar to that of India.

The future direction for this discussion could be to extrapolate the results across other countries and identify synergies that can offset some of the factors causing the performance to degrade due to higher liquidity. This extrapolation could also cover the geographical and economic limiting factors that arise within a particular country, India, in this case. A future study could further be extended to define the optimal liquidity balance and profitability for long-term stable banking with more data and details. Such a study could also pave the way to explore other neighboring regulations, such as Capital Requirements Regulation

and Capital Requirements Directive, and holistically assess the impact of multiple rules on a bank's ecosystem (profitability included).

Given the current macroeconomic conditions, liquidity regulations and norms have undoubtedly come a long way in preserving the practice of ethical banking. However, the optimum synergy is yet to be attained for the resiliency and sustainable effectiveness of achieving a harmonized balance between profitability and impounding regulations. The policy would require further statistics and contextualized industry analysis to create a standard protocol for the industry. Further sub-banking clusters would have to be formed to develop a sustainable equation that eventually addresses and champions the core principle of liquidity regulations to ensure liquidity but not at a steep cost to profitability.

**Author Contributions:** Conceptualization, S.R. and R.G.; methodology, V.M.B.; software, V.M.B.; validation, S.R. and R.G. and V.M.B.; formal analysis, V.M.B.; investigation, S.R.; resources, V.M.B.; data curation, R.G.; writing—original draft preparation, A.V.S.; writing—review and editing, V.M.B. and A.V.S.; visualization, S.R. and R.G.; supervision, S.R. and R.G.; project administration, S.R.; funding acquisition, S.R., R.G., V.M.B. and A.V.S. All authors have read and agreed to the published version of the manuscript.

**Funding:** This research received no external funding.

**Institutional Review Board Statement:** Not applicable.

**Informed Consent Statement:** Not applicable.

**Data Availability Statement:** Data is available and procured from CMIE Prowess database and the official website of the Reserve Bank of India.

**Conflicts of Interest:** The authors declare no conflict of interest.

## Abbreviations

| | |
|---|---|
| BCBS | Basel Committee on Banking Supervision |
| LCR | Liquidity Coverage Ratio |
| NSFR | Net Stable Funding Ratio |
| HQLAs | High-Quality Liquid Assets |
| NPAs | Non-performing Assets |
| NIMs | Net Interest Margins |
| LoLR | Lender of Last Resort |
| ROA | Return on Assets |

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
