# Peer review of "Impact of Liquidity Coverage Ratio on Performance of Select Indian Banks"

_jrfm, doi:10.3390/jrfm15050226_

Round 1

Reviewer 1 Report

REVIEW OF SUBMISSION TO JOURNAL OF RISK AND FINANCIAL MANAGEMENT – JFRM – 1685129 - “IMPACT OF LIQUIDITY COVERAGE RATIO OF PERFORMANCE OF SELECT INDIAN BANKS”

Summary of the paper

This paper uses data from India, to investigate the impact on bank profitability of Liquidity Coverage Ratios (LCR) and Non-performing Assets (NPA).  Two hypotheses are developed; both are articulated as bi-directional.  The first hypothesis (H1) postulates an association between bank performance and LCR.  The second hypothesis (H2) conjectures that LCR have an association with NPA.

The hypotheses are tested via Generalised Method of Moments, using panel data.  The first set of models, used to test H1, has earnings as the dependent variable.  Two earnings constructs are used: Return on Assets (ROA) and Net Interest Margin (NIM).  The independent variables include one-year lagged observation of the dependent variable, a measure of bank liquidity and controls.  Two measures of bank liquidity are used, in alternative specifications: LCR and LCR_p.  The second set of models, used to test H2, have NPA as the dependent variables.  The independent variables are analogous to their counterparts in the first set of regressions.  The sample comprises 31 Indian banks over the investigation period 2010-2019.

The results support the hypotheses.  However, they also suggest that the direction of association between LCR and the two dependent variables depends on how LCR is measured.  The paper reports positive autocorrelation in both the profitability and NPA of Indian banks.

Critical review

Introduction and motivation

The paper lacks an introduction.  The section titled “Introduction” (lucidly) argues a negative association between bank profitability and LCR.  Hence, this discussion belongs in the hypothesis development.

The authors argue convincing motivations.  The academic motivation stems from the dearth of evidence (particularly Indian) on how bank liquidity regulations impact their financial performance.  A business motivation is to inform bank regulators of possible economic consequences of the liquidity restrictions they impose on banks.

Literature review

The content of the literature review is sound.  However, the structure is poor.  I suggest the authors re-structure this section as follows.  They should commence with discussion of the rationale for subjecting financial institutions to liquidity regulations.  They should continue to discuss evidence that the immediate effects, on bank profitability, may be positive.  The next step should be to review evidence that these positive effects are likely to be short-lived and suggestions of possible reasons. The evidence from Tanzania, Indonesia and Europe would be relevant for this discussion.  The theme of liquid assets having a high opportunity cost is acknowledged but warrants more emphasis in this section.

Institutional environment of India

The authors should argue why the Indian experience may inform international readers.  There is some discussion of this nature, in the paragraph straddling pp.15 and 16.  However, elaboration is warranted.  It would be suitable to devote a section of the paper, to this theme.

Hypothesis development

The argumentation preceding H1 is convincing.  However, it is consistent with the association following an “inverted U”.  The hypothesis should be amended accordingly.  If the authors follow this suggestion, they will need to alter the methodology accordingly.  They may consider adding the square of the liquidity metrics, as an independent variable (Boubakri, Chen, El Ghoul, Geudhami and Nash, 2020).

H2 is not argued lucidly.  Most of the discussion argues a negative association between bank profitability and NPA.  This is a truism.  The argumentation should focus on the association between NPA and LCR.  Some discussion of this nature is provided, in the penultimate paragraph of Section 2.  Elaboration is needed. (e.g., does imposition of LCR constrain banks’ lending practices?)

Research methodology

Use of GMM is a strength of this paper.  The authors should explain its advantages more clearly.  GMM permits inclusion, as an independent variable, lagged observation of the dependent variable.  This models the dynamic evolution of the dependent variable and overcomes the problem that Ordinary Least Squares requires the independent variables to be non-stochastic.  These issues are discussed lucidly in Akbar, Poletti-Hughes, El-Faitouri and Zulfiqar, 2016).

The discussion should provide economic rationale for the controls.  Some coverage of this nature is presented on p.4 (e.g., the role of demographic and industry factors in explaining impacts of liquidity).  Elaboration would be appropriate.

Sample selection

It seems that the final sample comprises 310 bank-year observations.  Is this correct?  Alternatively, were some observations deleted, due to missing data?

Data collection

I have no criticisms of this dimension of the paper.

Descriptive statistics

The discussion accompanying Table 2 should include possible economic interpretations for the significant correlations.  Some suggestions follow.  The positive correlation (p<0.05) between ROA and NIM indicates that these are both constructs of earnings.  The negative correlation between ROA and size (p<0.05) may be due to Indian banks suffering from diseconomies of scale.

Empirical results

H1 is supported.  However, the results suggest that the direction of association between bank earnings and LCR depends on how the latter is measured.  This result is noted but not interpreted, in the economic context.  Similarly, H2 is supported.  However, the results suggest that the direction of association between NPA and LCR depends on how the latter is measured.  This result is noted but not interpreted, in the economic context. 

I commend the regression and endogeneity diagnoses. 

Conclusions

The authors conclude that the results suggest that imposing liquidity constraints on banks causes profit deterioration.  I believe that this conclusion is unwarranted, in light of the aforementioned conflicting evidence.

Presentation

The presentation of the tables could be improved.  Table 1 lists the two measures of liquidity, LCR and LCR­_p.  These variables should also be defined, quantitatively.  The tables presenting the descriptive statistics and empirical results should state the numbers of observations used.  Univariate descriptive statistics should be reported in a separate table from bivariate correlations.  More univariate statistics should be reported, such as medians and quartiles.

Colloquial language has been inappropriately used, in some sections.  Examples follows. On p.3, there is reference to “trading one devil for another” and “the story does not end”.  On p.17, the authors state that “norms have undoubtedly come a long way”.

Recommendation

I recommend rejection from Journal of Risk and Financial Management.

Suggestions for the authors

I encourage you to continue working on the paper.  It has solid foundations and potential.  Submission to any journal was premature.  I suggest you deliver at least three conference / seminar presentations.  Synthesise the feedback.  Modify the paper to address the criticisms (from both the presentations, my review and the reviews of the other referees), to the extent you regard them as valid.  Defer any consideration of journal submission, until you have taken these steps.

References, not cited in the manuscript

Akbar, S., J. Poletti-Hughes, R. El-Faitouri and S. Zulfiqar, 2016, “More on the Relationship between Corporate Governance and Firm Performance in the UK: Evidence from the Application of Generalised Method of Moments Estimation”, Research in International Business and Finance 38, 417-429.

Boubakri, N., R. Chen, S. El Ghoul, O. Geudhami and R. Nash, 2020, “State Ownership and Stock Liquidity: Evidence from Privatisation”, Journal of Corporate Finance 65, 1-26.

Author Response

Reviewer comments

Moderate English changes required

Authors Response

Apologies for the inconvenience caused. We have taken professional help from Grammarly services to improve the language

Reviewer comments

Does the introduction provide sufficient background and include all relevant references? (Must be improved)

Authors Response

Thanks for the feedback. It has helped us to structure our paper better. We have taken the reviewer’s suggestion to change the section by removing the discussion that supports hypothesis building and further develops the topic’s introduction.

Reviewer comments

Is the research design appropriate? (Must be improved)

Authors Response

Thanks for the valuable suggestion authors have considered it for the betterment of the paper and have worked on the data section

Reviewer comments

Are the methods adequately described? (Can be improved)

Authors Response

Thanks for the valuable suggestion authors have considered it for the betterment of the paper and have worked on the methodology section.

Reviewer comments

Are the results clearly presented? (Can be improved)

Authors Response

Thanks for the valuable suggestion authors have considered it for the betterment of the paper and have worked on the results section

Reviewer comments

Are the conclusions supported by the results? (Must be improved)

Authors Response

Thanks for the valuable suggestion. we have amended the conclusions to ensure they are in line with the results presented

Reviewer comments

The paper lacks an introduction. The section titled “Introduction” (lucidly) argues a negative association between bank profitability and LCR. Hence, this discussion belongs in the hypothesis development.

The authors argue convincing motivations. The academic motivation stems from the dearth of evidence (particularly Indian) on how bank liquidity regulations impact their financial performance. A business motivation is to inform bank regulators of possible economic consequences of the liquidity restrictions they impose on banks.

Authors Response

Thanks a lot for your suggestion. The authors have re-written the section to ensure that it provides sufficient background on the topic to set the context for the study. We have analyzed the diverse viewpoints that exist on the topic to identify the existing gaps, basis which we have formulated the objectives of the study and further elaborated on the contribution of the study.

We have removed the section that points toward the hypothesis building as per the reviewer’s suggestion.

Reviewer comments

The content of the literature review is sound. However, the structure is poor. I suggest the authors re-structure this section as follows. They should commence with discussion of the rationale for subjecting financial institutions to liquidity regulations. They should continue to discuss evidence that the immediate effects, on bank profitability, may be positive. The next step should be to review evidence that these positive effects are likely to be short-lived and suggestions of possible reasons. The evidence from Tanzania, Indonesia and Europe would be relevant for this discussion. The theme of liquid assets having a high opportunity cost is acknowledged but warrants more emphasis in this section.

Authors Response

Thanks for your valuable feedback. The authors have made the changes to the section as per the reviewer’s suggestions.

We have added the discussion on the need for liquidity. The same is available  from Pg 4 to 6

We have also re-structured the literature on Liquidity and Banks profitability in the following manner:

We start by providing the rationale for liquidity regulations. Then move on to discuss the impact of these regulations on banks’ profitability. In this, we begin with the literature that points towards the positive impact of the regulations on banks. Gradually we show how these positive impacts are not a permanent phenomenon, and banks start witnessing losses. In this section, we also talk about how the opportunity cost of holding additional liquid assets gradually starts outweighing the relatively low returns on the assets.

Then towards the end, we explain the plausible reasons for the gradual decline in banks’ profits. The changes are available on Page 6 and 7 of the paper

Reviewer comments

The authors should argue why the Indian experience may inform international readers. There is some discussion of this nature, in the paragraph straddling pp.15 and 16. However, elaboration is warranted. It would be suitable to devote a section of the paper, to this theme.

Authors Response

Thanks for the valuable suggestion. As per the reviewer’s suggestion, the authors have now added a separate section that talks about how the study conducted with particular reference to India would contribute to the existing body of knowledge. The section is available on Page 3 of the paper

Reviewer comments

The argumentation preceding H1 is convincing. However, it is consistent with the association following an “inverted U”. The hypothesis should be amended accordingly. If the authors follow this suggestion, they will need to alter the methodology accordingly. They may consider adding the square of the liquidity metrics, as an independent variable (Boubakri, Chen, El Ghoul, Geudhami and Nash, 2020).

H2 is not argued lucidly. Most of the discussion argues a negative association between bank profitability and NPA. This is a truism. The argumentation should focus on the association between NPA and LCR. Some discussion of this nature is provided, in the penultimate paragraph of Section 2. Elaboration is needed. (e.g., does imposition of LCR constrain banks’ lending practices?)

Authors Response

Thanks for your kind suggestion. The authors have now changed the hypothesis as per the reviewer’s suggestion and have updated the methodology accordingly. The changes are available on Page 7 of the paper

The authors have considered the quadratic terms of LCR and LCR_P and presented the results in Tables 4,5, and 6. the results obtained from the quadratic term show significance in one (model 4) out of the six models tested.

In line with the reviewers’ suggestion, we have re-written the argumentation for H2 by focussing on how the LCR impacts banks’ risk-taking and lending behaviour. The changes are available on Page 8 of the paper

Reviewer comments

Use of GMM is a strength of this paper. The authors should explain its advantages more clearly. GMM permits inclusion, as an independent variable, lagged observation of the dependent variable. This models the dynamic evolution of the dependent variable and overcomes the problem that Ordinary Least Squares requires the independent variables to be non-stochastic. These issues are discussed lucidly in Akbar, Poletti-Hughes, El-Faitouri and Zulfiqar, 2016).

The discussion should provide economic rationale for the controls. Some coverage of this nature is presented on p.4 (e.g., the role of demographic and industry factors in explaining impacts of liquidity). Elaboration would be appropriate.

Authors Response

Thanks for pointing out the issue, by focussing on the issue helped the authors to improve the quality of the paper. Now that the authors have considered the recommendations and using the citation support suggested, we improved the section reflected in dynamic panel data results using the track changes mode.

We have now provided the economic rationale for adding the controls to the study in the research methodology section. We thank the reviewer for the valuable suggestions that helped us improve the paper’s quality.

Reviewer comments

It seems that the final sample comprises 310 bank-year observations. Is this correct? Alternatively, were some observations deleted due to missing data?

Authors Response

Yes, the final sample comprises 310 observations for the study conducted. To bring to the reviewer’s notice, 34 private and public sector banks in India were initially considered for the study due to a lack of data availability. However, three banks were dropped from the sample to fit the study’s balanced panel, and the final sample was set at thirty-one banks. The same has been incorporated in the explanation of sources of the data section

Reviewer comments

I have no criticisms of this dimension of the paper.

Authors Response

Thanks for the encouragement

Reviewer comments

The discussion accompanying Table 2 should include possible economic interpretations for the significant correlations. Some suggestions follow. The positive correlation (p<0.05) between ROA and NIM indicates that these are both constructs of earnings. The negative correlation between ROA and size (p<0.05) may be due to Indian banks suffering from diseconomies of scale.

Authors Response

Thank you for bringing this to our attention. Our authors addressed the issue and attempted to incorporate an appropriate explanation of the correlation between the variables and what the relationship implies for the banks in India. The explanation for the same is present in the section Descriptive and correlation matrix.

Reviewer comments

H1 is supported. However, the results suggest that the direction of association between bank earnings and LCR depends on how the latter is measured. This result is noted but not interpreted, in the economic context. Similarly, H2 is supported. However, the results suggest that the direction of association between NPA and LCR depends on how the latter is measured. This result is noted but not interpreted, in the economic context.  I commend the regression and endogeneity diagnoses.

Authors Response

Thanks for the valuable suggestion. The authors have now added the relevant economic interpretation of the statistical results for all the significant findings of the study. The changes can be seen in the Dynamic panel data results section of the paper.

Reviewer comments

The authors conclude that the results suggest that imposing liquidity constraints on banks causes profit deterioration. I believe that this conclusion is unwarranted, in light of the aforementioned conflicting evidence.

Authors Response

Apologies for the possible confusion here. The study results indicate that with the increase in the LCR levels, banks' Net interest Margins (One of the measures of profitability) decrease. Hence we conclude that LCR and bank profitability have an inverse relationship

Reviewer comments

The presentation of the tables could be improved. Table 1 lists the two measures of liquidity, LCR and LCR­_p. These variables should also be defined, quantitatively. The tables presenting the descriptive statistics and empirical results should state the numbers of observations used. Univariate descriptive statistics should be reported in a separate table from bivariate correlations. More univariate statistics should be reported, such as medians and quartiles.

Colloquial language has been inappropriately used, in some sections. Examples follows. On p.3, there is reference to “trading one devil for another” and “the story does not end”. On p.17, the authors state that “norms have undoubtedly come a long way”.

Authors Response

Thanks for the valuable suggestion. Now authors have run the univariate statistics and reported the same in Table 3. Along with the median and quartile range, the authors have also mentioned variance, skewness, and kurtosis for the betterment of the paper.

Apologies for the inconvenience. The authors have now amended all the relevant parts where the colloquial language was used.

Reviewer 2 Report

Thank you for giving me the opportunity to review this paper. The paper examines the Liquidity Coverage Ratio (LCR) impact on the profitability and Non-Performing Assets (NPAs) of Indian banks using annual data from 2010 to 2019. However, there are some point need to be clarified. Hence, I proposed it for major revision:

  1. Please provide table of acronyms. There are too many acronyms in the manuscripts. Please revised and delete all less used acronyms.
  2. Please unify the terminology such as this paper, this study, this research or this article as either one of them.
  3. Please ensure that the abstract has the following elements: 1-2 sentences on the context and the need for the study; 1-2 sentences on the methodology; the majority of the abstract on the actual results of the study; 1-2 sentences on key conclusions and recommendations. It should be significantly revised to make sure the alignment between the title and study objective, results and contribution.
  4. The intro and lit review provide a lot of background on the topic at hand; however, in this manuscript, these 2 sections are somehow general. I think the authors could be further streamlined to be useful to set up the context: what is important on this topic, what research gaps exist, and what will this paper fill theoretically, empirically, and methodologically?
  5. The Literature review is comprehensive to demonstrate the understanding of the background studies. In general, the author should present the specific debate for your study. I would suggest the author to enhance your theoretical discussion and arrives your debate or argument. The author should update more latest work: For example: “Comparing world regional sustainable supply chain finance using big data analytics: a bibliometric analysis - https://doi.org/10.1108/IMDS-09-2020-0521”, “Challenges and trends in sustainable corporate finance: A bibliometric systematic review - https://doi.org/10.3390/jrfm13110264”, “A novel fuzzy credit risk assessment decision support system based on the python web framework - https://doi.org/10.1080/21681015.2020.1772385”, “Unreliable EPQ model with variable demand under two-tier credit financing - https://doi.org/10.1080/21681015.2020.1815877”.
  6. Where does the data come from? What idea or principle is behind the data? When and where is the data collected? Why do the authors choose the 10 years period? More detailed explanation about the data should be added.
  7. It was better to write the features of your method, the function of your method in the analysis. You should compare with conventional or standard method. A detailed explanation of why the method proposed by the author is better than other methods should be added.
  8. The 6 models should be explained in more detail.
  9. As to the data: Yet, some description of the rationale should be added in detail. Is it a practical case?
  10. The findings are interesting but need more explanation. The graphics and tables cannot speak for themselves. Rather, the author needs to describe what the findings show and explain/analyze, why and how they are important and pertain to the research questions/literature gaps.
  11. The Implication is needed for you to clarify your contribution to academic and practices. Hence, I would suggest you to place “implication section” separately. This implication has to address back to your industrial or case background as well as your theoretical background.
  12. Please make sure your conclusions' section underscore the scientific value added of your paper, and/or the applicability of your findings/results, as indicated previously. Basically, you should enhance your findings, limitations, underscore the scientific value added of your paper, and/or the applicability of your contributions/shortages and future study in this session.
  13. The reference and citation format need to be revised.

Author Response

Reviewer comments

Does the introduction provide sufficient background and include all relevant references? (Can be improved)

Authors Response

Thanks a lot for your suggestion. The authors have re-written the introduction section to ensure that it provides sufficient background on the topic to set the context for the study. We have analysed the diverse viewpoints that exist on the topic to identify the existing gaps, basis which we have formulated the objectives of the study and accordingly elaborated on the contribution of the study

Reviewer comments

Is the research design appropriate? (Can be improved)

Authors Response

Thanks for the valuable suggestion authors have considered it for the betterment of the paper and have worked on the data section

Reviewer comments

Are the methods adequately described? (Can be improved)

Authors Response

Thanks for the valuable suggestion authors have considered it for the betterment of the paper and have worked on the methodology section.

Reviewer comments

Are the results clearly presented? (Can be improved)

Authors Response

Thanks for the valuable suggestion authors have considered it for the betterment of the paper and have worked on the results section

Reviewer comments

Are the conclusions supported by the results? (Can be improved)

Authors Response

Thanks for the valuable suggestion. we have amended the conclusions to ensure they are in line with the results presented

Reviewer comments

Please provide table of acronyms. There are too many acronyms in the manuscripts. Please revised and delete all less used acronyms.

Authors Response

Apologies for the inconvenience. As per the reviewer’s suggestion, the authors have added a table for acronyms for the terms repeatedly used in the manuscript. We have also deleted all the less-used acronyms from the paper. The list of acronyms is available on Appendix of the manuscript.

Reviewer comments

Please unify the terminology such as this paper, this study, this research or this article as either one of them.

Authors Response

Apologies for the inconvenience. As per the reviewer’s suggestion, we have now unified the terminology across the article– This Paper, this study, and this research.

Reviewer comments

Please ensure that the abstract has the following elements: 1-2 sentences on the context and the need for the study; 1-2 sentences on the methodology; the majority of the abstract on the actual results of the study; 1-2 sentences on key conclusions and recommendations. It should be significantly revised to make sure the alignment between the title and study objective, results and contribution.

Authors Response

Thanks a lot for the suggestion; considering the reviewer’s feedback, the authors re-visited the abstract and found that it did not highlight the implication and contribution of the study. Accordingly, the abstract section has now been updated in the paper.

Reviewer comments

The intro and lit review provide a lot of background on the topic at hand; however, in this manuscript, these 2 sections are somehow general. I think the authors could be further streamlined to be useful to set up the context: what is important on this topic, what research gaps exist, and what will this paper fill theoretically, empirically, and methodologically?

Authors Response

Thanks a lot for your suggestion. The authors have re-written the literarture and introduction sections to ensure that it provides sufficient background on the topic to set the context for the study. We have analysed the diverse viewpoints/literature that exists for the topic to identify the existing gaps, basis of which we have formulated the objectives of the study and elaborated on the contribution of the study

Reviewer comments

The Literature review is comprehensive to demonstrate the understanding of the background studies. In general, the author should present the specific debate for your study. I would suggest the author to enhance your theoretical discussion and arrives your debate or argument. The author should update more latest work: For example: “Comparing world regional sustainable supply chain finance using big data analytics: a bibliometric analysis - https://doi.org/10.1108/IMDS-09-2020-0521”, “Challenges and trends in sustainable corporate finance: A bibliometric systematic review - https://doi.org/10.3390/jrfm13110264”, “A novel fuzzy credit risk assessment decision support system based on the python web framework - https://doi.org/10.1080/21681015.2020.1772385”, “Unreliable EPQ model with variable demand under two-tier credit financing - https://doi.org/10.1080/21681015.2020.1815877”.

Authors Response

Thanks for your feedback. The authors have made the changes to the sections as per the reviewer’s suggestions. We have enhanced our literature to provide sufficient theoretical background to support our argument and build our hypothesis. We have also added the suggested latest work to the paper to make it more relevant to the current banking environment

Reviewer comments

Where does the data come from? What idea or principle is behind the data? When and where is the data collected? Why do the authors choose the 10 years period? More detailed explanation about the data should be added.

Authors Response

For diverse reasons, including the 2008 global financial crisis, the Indian banking industry is undergoing major restructuring (Khalil and Slimene, 2021). Furthermore, the merger of government-induced public sector banks in 2019 has caused significant upheaval in the Indian banking industry. As a result, bank data was collected until 2019. A detailed explanation of the data collation and justification for choosing the time is explained in the section sources of the data.

Reviewer comments

It was better to write the features of your method, the function of your method in the analysis. You should compare with conventional or standard method. A detailed explanation of why the method proposed by the author is better than other methods should be added.

Authors Response

Thanks for your valuable recommendations. Authors have now considered the suggestions, and the required explanation is provided in the section Dynamic panel data results with the citation help of Arellano and Bover (1995), Blundell and Bond (1998), and  Akbar et al. (2016)

Reviewer comments

The 6 models should be explained in more detail. As to the data: Yet, some description of the rationale should be added in detail. Is it a practical case?

Authors Response

The recommendation is considered and all the 6 models are explained in detail. We extend our thanks to the reviewers for the valuable suggestions that helped us improve the paper’s quality.

Reviewer comments

The findings are interesting but need more explanation. The graphics and tables cannot speak for themselves. Rather, the author needs to describe what the findings show and explain/analyze, why and how they are important and pertain to the research questions/literature gaps.

Authors Response

Thanks for the valuable suggestion. The authors have now added the relevant economic interpretation of the statistical results for all the study's significant findings. The changes can be seen in the Dynamic panel data results section of the paper.

Reviewer comments

The implication is needed for you to clarify your contribution to academic and practices. Hence, I would suggest you to place “implication section” separately. This implication has to address back to your industrial or case background as well as your theoretical background

Authors Response

Thanks for the valuable feedback. The authors have now added a separate section to discuss the result implications. The section has been added as a sub-section in the discussion part of the paper.

Reviewer comments

Please make sure your conclusions’ section underscore the scientific value added of your paper, and/or the applicability of your findings/results, as indicated previously. Basically, you should enhance your findings, limitations, underscore the scientific value added of your paper, and/or the applicability of your contributions/shortages and future study in this session.

Authors Response

Thanks for the valuable suggestion. The authors have re-worked on the conclusion section to highlight important findings, contribution, applicability and limitation of the study.

Reviewer comments

The reference and citation format need to be revised.

Authors Response

Apologies for the inconvenience, we have re - checked the reference and citation format to make sure everything is in line with the journal formatting.

Reviewer 3 Report

The current paper 'Impact of Liquidity Coverage Ratio on Performance of Select Indian Banks' examines the impact of Liquidity Coverage Ratio on the profitability and Non-Performing Assets in a contextual scenario of Indian banks. I have few suggestions/ comments for the authors

1) The objective of the study (To study the impact of Liquidity regulations on the performance of Indian Banks with particular reference to short term liquidity AND to explore  the impact of Liquidity regulations on NPA levels of Indian Banks) . How will you define the proxy for liquidity regulations?

2) One of the potential novelty of this paper (highlighted by the authors) is to examine the effect of liquidity regulations on the performance of Indian banks for pre-and post-implementation periods. For the reader, it is quite challenging to determine how authors compared pre and post regulation period? What is the significant difference/ change? Is the context or the data period (or both) makes this study novel?

3) The introduction and literature review sections must be improved by adding more literature citations.

4) The paper should go through a thorough language check for errors and misprints (e.g., In the first paragraph of conceptual model of the study,  LCR is liquidity coverage ratio or liquidity capital ratio?) 

Author Response

Reviewer comments

Moderate English changes required

Authors Response

Apologies for the inconvenience caused. We have taken professional help from Grammarly services to improve the language

Reviewer comments

Does the introduction provide sufficient background and include all relevant references? (Must be improved)

Authors Response

Thanks a lot for your suggestion. The authors have re-written the sections to ensure that it provides sufficient background on the topic to set the context for the study. We have analyzed the diverse viewpoints/literature that exists for the topic to identify the existing gaps, basis of which we have formulated the objectives of the study and elaborated on the contribution of the study.

Reviewer comments

Is the research design appropriate? (Must be improved)

Authors Response

Thanks for the valuable suggestion authors have considered it for the betterment of the paper and have worked on the data section

Reviewer comments

Are the methods adequately described? (Can be improved)

Authors Response

Thanks for the valuable suggestion authors have considered it for the betterment of the paper and have worked on the methodology section.

Reviewer comments

Are the results clearly presented? (Must be improved)

Authors Response

Thanks for the valuable suggestion authors have considered it for the betterment of the paper and have worked on the results section

Reviewer comments

Are the conclusions supported by the results? (Can be improved)

Authors Response

Thanks for the valuable suggestion. we have amended the conclusions to ensure they are in line with the results presented

Reviewer comments

The objective of the study (To study the impact of Liquidity regulations on the performance of Indian Banks with particular reference to short term liquidity AND to explore the impact of Liquidity regulations on NPA levels of Indian Banks) . How will you define the proxy for liquidity regulations?

The paper uses the Liquidity coverage ratio as the proxy for liquidity regulation. It is one of the two ratios that have been introduced under the new liquidity regulations by the Basel committee and measures the bank’s capability to fulfill its obligations for thirty days if crisis circumstances arise

 One of the potential novelty of this paper (highlighted by the authors) is to examine the effect of liquidity regulations on the performance of Indian banks for pre-and post-implementation periods. For the reader, it is quite challenging to determine how authors compared pre and post regulation period? What is the significant difference/ change? Is the context or the data period (or both) makes this study novel?

Authors Response

Apologies for the possible confusion here. Kindly note that the comparison of the pre and post implementation era is not the paper’s objective. We are examining the impact of LCR on the profitability (Net Interest Margins and Return on Assets) and Non-Performing Assets of the Indian banks for the period 2010 to 2019, which covers both pre and post-implementation period as regulations got implemented in India only post 2015. We have accordingly modified the wordings in the paper to avoid this confusion. The changes can be seen in second last para on page 3

Reviewer comments

The introduction and literature review sections must be improved by adding more literature citations

Authors Response

Thanks for the valuable suggestion. We have further elaborated the introduction and literature section to set the context for the study. As per the reviewer’s suggestion, we have added more literature citations in the paper to strengthen the theoretical background

Reviewer comments

The paper should go through a thorough language check for errors and misprints (e.g., In the first paragraph of conceptual model of the study, LCR is liquidity coverage ratio or liquidity capital ratio?)

Authors Response

Extremely apologize for the inconvenience caused, and thanks for pointing out our mistake. Now the error is rectified, and the word is coverage, not capital; we have corrected our mistake and thoroughly checked for the language errors recommended by the reviewers.

Round 2

Author Response

1) English language and style are fine/minor spell check required

Authors response- Apologies for the inconvenience. We have run through a thorough spell-check to ensure correctness.

2) Does the introduction provide sufficient background and include all relevant references? - YES

Authors response- Thanks for the kind and encouraging words.

3) Are all the cited references relevant to the research? – YES

Authors response- Thanks for the kind and encouraging words.

4) Is the research design appropriate? – YES

Authors response- Thanks for the kind and encouraging words.

5) Are the methods adequately described? (Can be improved)

Authors response- Thanks for the recommendation made, the authors have in the methodology and results section and highlighted the same with the yellow.

6) Are the results clearly presented? (Can be improved)

Authors response- Thanks for the valuable suggestion authors have considered it for the betterment of the paper and have worked on the results section by incorporating all the suggestions given by the reviewers.

7) Are the conclusions supported by the results? (Must be improved)

Authors response- Thanks for the valuable suggestion. We have made the necessary changes to ensure that conclusions are adequately supported by the results with due justification.

8) I have one remaining comment. The authors have combined the literature review and hypothesis development, within the same section of the paper. This is a legitimate strategy. However, this section should be titled “Literature review and hypothesis development”.

Authors response- Thanks a lot for the valuable suggestion. We have now changed the title of the section to “Literature review and hypothesis development”. Same can be seen on Page 4 of the manuscript now.

9) The paper now contains adequate discussion of the institutional environment of India. I accept that the authors have chosen to integrate this content throughout the paper, rather than dedicating a section to this purpose.

Authors response- Thanks for the kind and encouraging words.

10) The authors have not implemented my suggestion, of articulating in H1, the expectation of an inverted U. However, they have convincingly argued a different case. The amended discussion suggests that over time, the impact on bank profitability, of imposed liquidity constraints, deteriorates. This is a completely separate issue from the association between bank profitability and the level of liquidity. Hence, the authors have appropriately articulated this hypothesis generally (the expectation of a non-linear association), rather than more specifically (the expectation of an inverted U).

The development of H2 has improved. However, it warrants more improvement. The authors discuss evidence from Europe, consistent with a positive association between banks’ nonperforming assets and the level of liquidity. The authors continue to note that evidence from Europe may not generalise to India. This is a salient observation. However, they should present a possible reason why the Indian situation is different. Apposite discussion is located within the final paragraph of S5.1; the paper notes that in India, high levels of non-performing assets may lead to moral hazard and other risky lending activities, ultimately causing deterioration in liquidity. To address this concern, I suggest the authors re-locate some of the discussion from S5.1 into their development of H2.

Authors response- Apologies for the confusion here. As per the reviewer’s suggestion, we start with literature that shows a positive impact of Liquidity regulations on performance of banks, gradually we move towards the literature that supports that the positive impact may not be a permanent phenomenon and then we explain how the profitability of banks starts declining if additional liquidity is infused. In order to bring out the concept more clearly we have added a concluding paragraph just before stating H1. The H1 has been accordingly updated to “LCR and bank profitability exhibit an inverted U-Shaped relationship.” Same can be seen on Page 7 of the manuscript.

Thanks a lot for the valuable suggestion. As per reviewer’ notes, we have now enhanced the discussion for H2 by elaborating on how the Indian situation is different from that of Europe when it comes to possible impact of LCR on NPAs of banks. Same can be seen on Page 8 of the manuscript.

11) The discussion accompanying Table 3 warrants improvement. The authors appropriately note that ROA has a negative skew whereas non-performing assets have a positive skew. Provide a possible economic interpretation. (e.g., a large percentage of the banks have incurred substantial losses, possibly due to the risky lending and moral hazard problems noted in S5.1. This could also explain why a relatively large percentage of the sample banks have high non-performing assets.)

Authors response- Thanks, a lot for the valuable suggestion. As per the reviewer’s suggestion, the authors have added relevant discussion on the Table 3 to elaborate on the results with due interpretations and justifications.

12) Throughout the paper, the authors claim that the results support neither a linear, nor a nonlinear association. This would be scientifically impossible! Everywhere this claim is made, it must be replaced with the claim that the results support neither a linear, nor a quadratic association.

A key result is that in Table 4, Model (1) reports a negative coefficient of LCR. Conversely, Model (2) displays a positive coefficient of LCR_p. This result is duly noted, in S4.2. However, more discussion is warranted. I suspect that the difference in coefficient sign reflects the different specifications of the two measures of liquidity. The authors should reflect on this and argue which result is contextually more credible. This affects their conclusions of the direction of association between bank profitability and liquidity

The aforementioned comment, regarding Table 4, also applies to Table 6. (i.e., the coefficient of LCR in Model (1) of Table 6 is positive. Conversely, the coefficient of LCR_p in Model (2) is negative.)

Authors response-

Extreme apologies for the mistake made by us, now we authors have replaced the non-linear term with quadratic term. We authors thank reviewers for pointing out and correcting us.

Thanks a lot for the valuable suggestion, as per reviewer’s notes, we have elaborated more on interpretation of results for Model 1 and 2 in light of different specifications of LCR and LCR_P. The results of both the models have same conclusion that liquidity has an adverse impact on the profitability of banks. We have explained and demonstrated this clearly now in the results section. Further, in section 5 of the paper- “Discussion and Implication of results”, we have discussed in length about the LCR results as since LCR has been calculated as per the BASEL guidelines and is contextually more credible measure as compared to LCR_P. We have also highlighted the same in the beginning of the Section 5. Apologies for not covering this explicitly in the previous manuscript.

Thanks a lot for the valuable suggestion, as per reviewer’s notes, we have elaborated more on interpretation of results for Model 5 and 6 in light of different specifications of LCR and LCR_P. The results of both the models have same conclusion that with an increase in the liquidity the NPAs are increasing as well. We have explained and demonstrated this clearly now in the results section. 

13) The authors should provide a within-text reference, to support the statistic quoted regarding the United States government’s bailout of Citibank.

My concerns regarding Table 1 have not been fully addressed. The authors have explained the economic interpretations of LCR and LCR_p. They should define these variables quantitatively, as they have defined return on assets. (i.e., how are the numerator and denominator of LCR and LCR_p specified?)

There is a critical omission from Equations (1) and (2), in S3.4. The squared terms, independent variables in the estimated models, are not listed in the equations. The presentation Table 3 needs to be improved. In S4.1, the authors discuss the means of NIM, ROA and NPA. These means should be reported in the table. Similarly, it is more conventional to report the upper and lower quartiles as separate columns.

In Tables 4, 5 and 6, the entries for LCR and LCR_p should also be highlighted. These are also independent variables of interest.

Authors response- Apologies, we have now added the reference for the static quoted and same can be seen on Page 1 of the manuscript.

Apologies, we have updated the table to reflect the quantitative calculation of the variables in Table 1 under the description head.

Extremely apologies for the omission of the squared terms in the econometric equations. We authors have now included the squared terms in the respective equations. We have also introduced the means in the table 3 and have highlighted the same in yellow. Also, the quartiles have been reported in separate columns- Upper and Lower quartiles. We have highlighted the column in yellow for easy reference.

Thanks for the valuable suggestion, we have highlighted the entries for LCR and LCR_P in tables 4,5 and 6 as per reviewer’s suggestion.

14) The authors have made substantial progress, in addressing the concerns broached in my report, regarding the previous version of the manuscript. However, notable concerns remain. Hence, I recommend that the authors be invited to re-submit the paper to Journal of Risk and Financial Management, after making major changes.

Authors response- Thanks a lot for the kind and encouraging words. We are really thankful for all the suggestions and guidance provided by the reviewers; it has been really helpful in improving the quality of our paper.

Reviewer 2 Report

This manuscript has been revised and followed the reviewers’ comments. Hence, I would like to recommend this manuscript to be accepted.

Author Response

1) English language and style are fine/minor spell check required

Authors response- Apologies for the inconvenience. We have run through a thorough spell check to ensure correctness.

2) Does the introduction provide sufficient background and include all relevant references? - YES

Authors response- Thanks for the kind and encouraging words.

3) Are all the cited references relevant to the research? – YES

Authors response- Thanks for the kind and encouraging words.

4) Is the research design appropriate? – YES

Authors response- Thanks for the kind and encouraging words.

5) Are the methods adequately described? - YES

Authors response- Thanks for the kind and encouraging words.

6) Are the results clearly presented? - YES

Authors response- Thanks for the kind and encouraging words.

7) Are the conclusions supported by the results? -YES

Authors response- Thanks for the kind and encouraging words.

Reviewer 3 Report

The authors have incorporated most of the suggestions/comments. 

Author Response

1) English language and style are fine/minor spell check required

Authors response- Apologies for the inconvenience. We have run through a thorough spell check to ensure correctness.

2) Does the introduction provide sufficient background and include all relevant references? - YES

Authors response- Thanks for the kind and encouraging words.

3) Are all the cited references relevant to the research? – YES

Authors response- Thanks for the kind and encouraging words.

4) Is the research design appropriate? (Can be improved)

Authors response- Considering the valuable suggestion made by the reviewers authors had improved the methodology section and highlighted the same in yellow

5) Are the methods adequately described? (Can be improved)

Authors response- Thanks for the recommendation made, the authors have in the methodology and results section and highlighted the same with the yellow.

6) Are the results clearly presented? (Can be improved)

Authors response- Thanks for the valuable suggestion authors have considered it for the betterment of the paper and have worked on the results section

7) Are the conclusions supported by the results? -YES

Authors response- Thanks for the kind and encouraging words.

Round 3

Reviewer 1 Report

18 May, 2022

REVIEW OF SUBMISSION TO JOURNAL OF RISK AND FINANCIAL MANAGEMENT – JFRM – 1685129 – ROUND 3 - “IMPACT OF LIQUIDITY COVERAGE RATIO OF PERFORMANCE OF SELECT INDIAN BANKS”

Note

The clarity of the paper is substantially improved, compared to previous versions.  This has enabled me to write a more considered critique.  Hence, I have reverted to the practice of preparing a comprehensive report, as I would for a first-round submission.

Summary of the paper

This paper investigates the impact, on the performance of Indian banks, of imposition of liquidity regulations.  The authors argue that the case, for subjecting banks to liquidity regulations, to avert market failure, is particularly strong in India.  The discuss information asymmetry between banks and depositors and incentives for banks to engage in moral hazard.  One mode of committing moral hazard would be to choose capital expenditure projects that are excessively risky.  There would be incentive to do so if banks were regulated via means other than liquidity controls, such as a Lender of Last Resort facility.

The paper develops two hypotheses.  The first hypothesis conjectures that the association between bank profitability and liquidity follows an inverted U.  The second hypothesis is non-directional, articulating an association between bank non-performing assets and liquidity.  These hypotheses are tested via General Method of Moments. The first hypothesis is tested via a model with bank profitability, alternatively measured as return on assets (ROA) and net interest margin, as the dependent variable.  The independent variables include a one-year lag of the dependent variable, bank liquidity (alternatively measured via two different ratios), the square of the bank liquidity metric and two control variables.  One of the liquidity metrics is a direct measure; the other is an inverse measure.  The investigation period is 2010-2019.  The final sample comprises a panel dataset of 340 bank-year observations.

The results, using net interest margin as the measure of profitability, support a monotonically negative association between bank profitability and liquidity.  The results using ROA do not support the same conclusion.  The tests of the second hypothesis support a monotonically positive association between non-performing assets and bank liquidity.

Critical review

The motivation and underlying theory and sound and clearly articulated.  The first hypothesis is developed lucidly.  The arguments preceding the second hypothesis are convincing.  However, they are consistent with a monotonically positive association between bank non-performing assets and liquidity, rather than an inverted U.  I suggest the authors amend this hypothesis, accordingly.

I recommend the authors delete all analysis, using ROA as the measure of profitability.  They provide lucid arguments that ROA may be difficult to interpret, as a performance metric for banks.  This position is generally accepted in the literature.  The following empirical findings in the paper support my suggestion.  Table 2 reports a negative correlation between net interest margin and ROA (p<0.10).  The results in Table 5, testing H1 using ROA as the dependent variable, are difficult to interpret.

I have two other principal suggestions, regarding the methodology.  Firstly, the authors should clarify in the methodology section that LCR is a direct measure of liquidity whereas LCR­­_p is an indirect measure.  The underlying reason is explained clearly, in the discussion of results.  The paper would be easier to follow, if this were explained earlier.  A corollary of this suggestion is that they should augment Table 2, to include the correlation between these two liquidity metrics.  I anticipate a negative correlation.  Secondly, if the authors choose to follow my suggestion regarding H2 (of a positive, monotonic association), they should delete the squared term altogether, as an independent variable in Equation (2).  The definition of LCR_p2 should be inserted into Table 1, containing the variable definitions.

I have no concerns regarding the sample selection; nor do I have any concerns about the data collection.

I have some concerns about the descriptive statistics.  Firstly, it is more conventional to report univariate statistics before bivariate statistics.  Hence, I suggest the authors reverse the order of Tables 2 and 3.  The second sentence in Section 4.1 relates to univariate descriptive statistics, rather than bivariate correlations.

My remaining concern about the empirical results relates to Table 6.  If the authors accept my criticism about H2, the results reported in this table will need to be re-estimated, without LCR­_p2 as an independent variable.  I believe that prima facie, these results support the conjecture of a monotonically positive association.  (i.e., the coefficient of the direct (inverse) liquidity metric is positive (negative) and significant (p<0.01).  Furthermore, the coefficients of the squared terms are uniformly insignificant.)

I have no other concerns about the presentation.  This aspect of the paper shows a marked improvement.

Recommendation

This paper is improving, with every iteration.  However, it is still not ready for publication.  I recommend that the authors be invited to re-submit the paper to Journal of Risk and Financial Management, after making major changes.